# Lifelong Learning of Video Diffusion Models From a Single Video Stream

## Abstract

This work demonstrates that training autoregressive video diffusion models from a single, continuous video stream is not only possible but can also be as effective as standard offline training approaches given the same number of gradient steps. Our demonstration further reveals that this main result can be achieved using experience replay that only retains a subset of the preceding video stream. We also contribute three new single video generative modeling datasets suitable for evaluating lifelong video model learning: *Lifelong Bouncing Balls*, *Lifelong 3D Maze*, and *Lifelong PLAICraft*. Each dataset contains over a million consecutive frames from an environment of increasing complexity.[1]

## 1. Introduction

There are a plethora of names – lifelong learning, continual learning, streaming inference – given to a central desideratum of artificial intelligence (AI) systems: the ability to learn from a single continuous autocorrelated stream of data. However named, our community has long sought models and algorithms that learn in a fundamentally human way; from birth to death, learning as we live.

Modern AI systems do not learn in this manner but instead rely on a effective compromise: stochastic gradient descent from data streams made up of independently and identically distributed (i.i.d.) samples. Language models (Touvron et al., 2023; Mukherjee et al., 2023), world models (Hafner et al., 2020; 2021), and video models (Harvey et al., 2022; Ho et al., 2022; Bar-Tal et al., 2024) are trained on random batches of short temporally correlated segments, an approach that preserves short-range autocorrelations while approximating i.i.d.-ness through permutations. While effective, this approximately i.i.d. learning paradigm does not

offer a satisfying mechanism for updating models when new data arrive. Existing model updating techniques that are often used in practice—training from a checkpoint on the union of old and new data (Ash & Adams, 2020), fine-tuning on just the new data (Xu et al., 2023), or training completely anew from scratch (Ren et al., 2021) – do not operate in the training regime from which humans learn and suffer from problems such as high computation cost and forgetting (Verwimp et al., 2024).

Alternatively, SGD on autocorrelated data streams is considered by some to be a viable candidate for human-like lifelong learning (Lillicrap et al., 2020). While there is a raft of work indicating that gradient-based learning on autocorrelated data is possible (Duchi et al., 2012; Johansson et al., 2010; Ram et al., 2009), folk wisdom maintains that this learning setup is hard, impractical, and prone to failure, especially for deep networks. Evidence of these beliefs can be found throughout the literature. The optimization community has developed numerous mechanisms for alleviating the effects of data stream temporal dependencies (Kowshik et al., 2021; Godichon-Baggioni et al., 2023; Chang & Shahrampour, 2022), suggesting the existence of problems with learning from autocorrelated data streams. Additionally, the Bayesian learning community has proposed numerous continual learning approaches through posterior updating (Bartlett & Wood, 2011; Broderick et al., 2013; Naesseth et al., 2019; Beronov et al., 2021), though these theoretically sound approaches lack practical scaled results. These bodies of work suggest a strong desire to avoid non-i.i.d. learning, making it reasonable to assume that lifelong learning of video models from a single autocorrelated video stream might be difficult if not impossible.

The primary contribution of this paper is simple yet profound. We provide an empirical proof-of-concept that diffusion-based video models can be learned in a lifelong manner from a single autocorrelated video stream while obtaining similar performance to standard (offline) i.i.d. training. We demonstrate successful lifelong learning even on highly nonstationary partially-observable 3D domains, despite (to the best of our knowledge) being the first demonstration of non-i.i.d. training of video diffusion models. Remarkably, successful lifelong learning does not require a complex setup. We find that a minimal set of established continual learning techniques, such as experience replay

---

[1]Anonymous Institution, Anonymous City, Anonymous Region, Anonymous Country. Correspondence to: Anonymous Author <anon.email@domain.com>.

Preliminary work. Under review by the International Conference on Machine Learning (ICML). Do not distribute.

[1]Code and datasets will be released on acceptance.

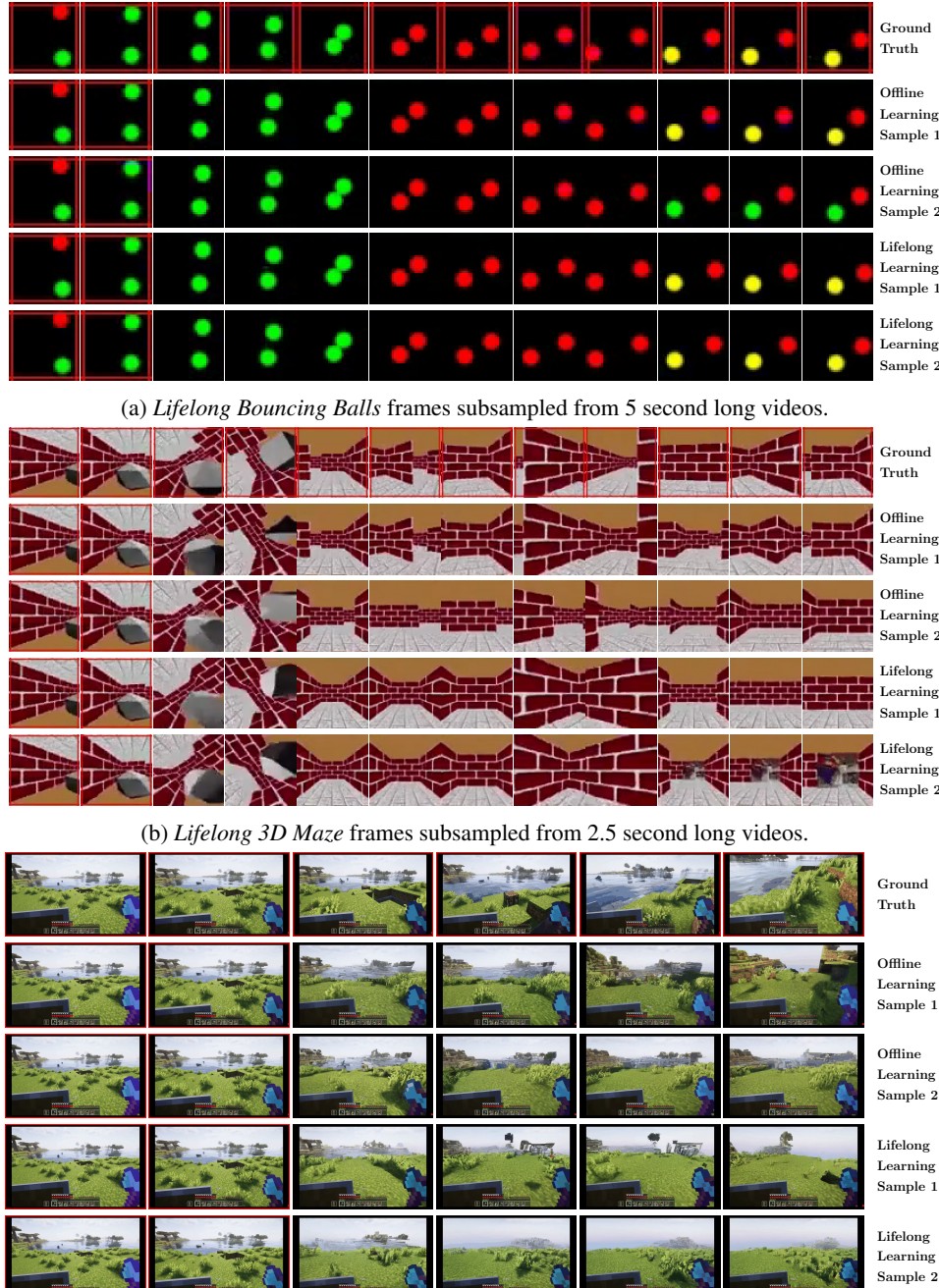

(a) *Lifelong Bouncing Balls* frames subsampled from 5 second long videos.

(b) *Lifelong 3D Maze* frames subsampled from 2.5 second long videos.

(c) *Lifelong PLAICraft* frames subsampled from 3 second long videos.

Figure 1. Ground truth video frames (1st row of each subfigure), offline learned models' generated videos (2nd and 3rd rows), and lifelong learned models' generated video frames (4th and 5th rows) for our datasets. The left two columns highlighted in red show the frame conditioned upon the model. Videos generated by lifelong learned models trained with experience replay are diverse, visually plausible, and indistinguishable from those of offline learned models in quality.

with limited memory, is sufficient to make lifelong- and i.i.d.-trained models qualitatively indistinguishable given the same numbers of gradient steps and batch size.

As a secondary contribution, we introduce three novel lifelong learning video datasets with varying levels of temporal correlation, data repetitiveness, rare events, and non-stationarity. Even under an academic computational budget, we observe stable learning and reliable short-range extrapolations on all three benchmarks. As video modeling is a component of world modeling, our findings have the potential to open up new life-like approaches to learning, planning, and control in embodied AI agents.

## 2. Background

**Video Diffusion Models** Video Diffusion Models (VDMs) are a class of generative models capable of synthesizing high-quality, temporally consistent videos (Voleti et al., 2022; Ho et al., 2022; Harvey et al., 2022; Höppe et al., 2022; Green et al., 2024; Blattmann et al., 2023b;a; Brooks et al., 2024). Rooted in the principles of denoising diffusion, video diffusion models progressively refine random noise into coherent video frames across a series of iterative denoising steps. Extending image diffusion models to videos requires capturing temporal dependencies between frames while preserving single-frame quality. These requirements present unique challenges, including the computational overhead of processing a large number of frames as well as developing architectures that can capture both spatial and temporal coherence efficiently. To handle long-duration videos, models learn the conditional probability of new frames given previously generated frames and then generate videos in an auto-regressive manner (Harvey et al., 2022; Deng et al., 2024).

**Lifelong Learning** The goal of lifelong or continual learning is enabling models to continuously learn from new data with minimal forgetting of what was learned before (De Lange et al., 2022; Wang et al., 2024; van de Ven et al., 2024; Yoo & Wood, 2022). Approaches for lifelong learning include using regularization to penalize changes to parts of the network that encode previously learned information (Kirkpatrick et al., 2017; Zenke et al., 2017; Li & Hoiem, 2017), improving the plasticity or the stability of the optimization algorithm (Dohare et al., 2024; Hess et al., 2023; Yoo et al., 2024), and enforcing the encoding of different tasks in minimally overlapping or orthogonal parts of the model (Rusu et al., 2016; Serra et al., 2018; Zeng et al., 2019). Another popular approach is replay, whereby the model revisits samples representative of past data along with the current training data (Robins, 1995). Such replayed samples can be obtained from generative models (Shin et al., 2017), but often they are sampled from a memory buffer containing past training data, an approach referred to as experience replay (Chaudhry et al., 2019; Rolnick et al., 2019; Buzzega et al., 2020; Arani et al., 2022).

Most work on neural network lifelong learning has focused on the highly simplified problem setting where a classification model is trained on a sequence of non-overlapping tasks, with each task seen only once. Often the model can train on each new task until convergence (referred to as the offline setting), although some works only allow a single pass over the data of each task (the online setting) (Aljundi et al., 2019; Chen et al., 2020). Beyond classification, lifelong learning research has explored the continual training of generative models, including generative adversarial networks, variational auto-encoders and diffusion models for image generation (Zhai et al., 2019; Lesort et al., 2019; Egorov et al., 2021; Smith et al., 2024). However, as far as we are aware, no work thus far has explored lifelong learning of video diffusion models. The lifelong learning community has also shown interest in moving beyond learning in a strictly task-based manner. To create data streams with more complex temporal correlations, blurry task boundaries (Bang et al., 2022; Moon et al., 2023) and repetition of previously seen concepts (Hemati et al., 2024) have been used. Benchmarks for lifelong learning have also been constructed based on data from autonomous driving (Verwimp et al., 2023), using image datasets collected through time (Bornschein et al., 2023) and by concatenating thousands of short videos (Carreira et al., 2024).

## 3. Datasets

Exploring the possibility of learning video models in a lifelong fashion requires (1) long continuous video streams and (2) varying levels of complexity to gauge challenges and limitations. As discussed in Section 2, existing video datasets usually consist of many short videos that may or may not be temporally related. To that end, we introduce new video datasets constructed from single video streams without semantic discontinuities. The three datasets – *Lifelong Bouncing Balls*, *Lifelong 3D Maze*, and *Lifelong PLAICraft* video datasets – vary in terms of complexity, stochasticity, temporal correlation, and degree of non-stationarity.

### 3.1. Lifelong Bouncing Balls

We introduce two versions of the *Lifelong Bouncing Balls* dataset: Version *O* for "original" and *C* for "changing".

*Lifelong Bouncing Balls (O)* contains 1 million 32x32 RGB video frames for training ($\sim$28 hours long at 10FPS) and another 1 million video frames for evaluation. The video contains two colored balls that deterministically bounce around in a 2D environment, colliding with boundaries and each other. Upon collision, each ball changes velocity according to the conservation of momentum and cycles through a repeating color sequence of "red" $\rightarrow$ "yellow" $\rightarrow$ "red" $\rightarrow$ "green". Since the balls' states are fully observable and their transition dynamics are deterministic, the future frames can be perfectly predicted. Solving *Lifelong Bouncing Balls (O)* requires learning the 2D environment's **deterministic** dynamics and retaining the balls' color transition histories from a **correlated** and **repetitive** video stream.

*Lifelong Bouncing Balls (C)* introduces non-stationarity on top of *Lifelong Bouncing Balls (O)*. While the balls' motion matches that of (O), the blue channel values of all ball colors increase over time at a constant rate (red, yellow, and green ball colors respectively become fuchsia, white, and aqua by the end of the video). The evaluation set contains frames

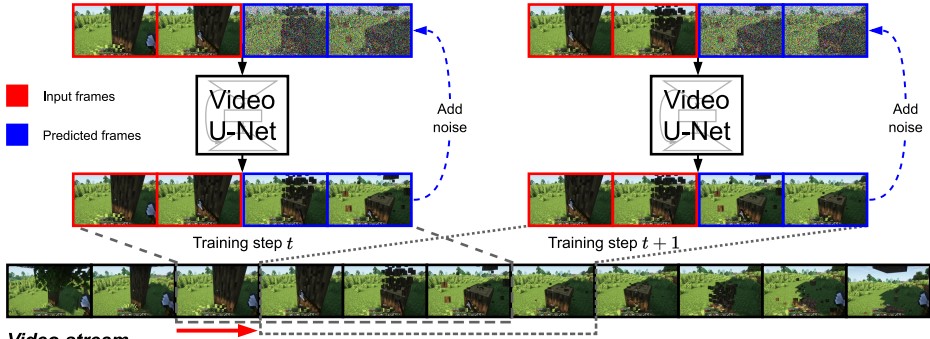

*Figure 2.* Video diffusion model lifelong learning on a single video stream with $K = 4$. At training step $t$, the model conditions on the frames in the first half of its context window (red) and learns to denoise the frames in the second half of its context window (blue). At training step $t + 1$, the model's context window shifts by one video frame, and the same procedure repeats indefinitely.

with balls of all previously observed colors. Solving *Lifelong Bouncing Balls (C)* requires learning the **deterministic** ball dynamics and color transitions from a **correlated** video stream where some details are **unrepeated**.

### 3.2. Lifelong 3D Maze

The *Lifelong 3D Maze* dataset contains 1 million 64x64 RGB video frames for training (∼14 hours long at 20FPS) and 100,000 video frames for evaluation. The video is a first-person view of an agent that navigates a randomly generated 3D maze. Whenever the agent solves a maze, the walls of the solved maze come down and the walls of an unseen maze rise, at which point the agent attempts to solve the new maze. The mazes contain various sparsely appearing objects, including polyhedral gray rocks that flip the agent upside down upon being touched. Since the maze states are partially observable and their transition dynamics are stochastic, the future frames cannot be perfectly predicted given the past frames. Solving *Lifelong 3D Maze* requires learning the first-person sensory inputs associated with navigating a **stochastically generated** 3D environment and modeling **infrequent events** from a largely **repetitive** and **correlated** video stream.

### 3.3. Lifelong PLAICraft

The *Lifelong PLAICraft* dataset contains 1.85 million 1280x768 RGB video frames for training (∼54 hours long at 10FPS) and 500,000 video frames for evaluation. To make training computationally feasible, we encode these video frames using the Stable Diffusion perceptual encoder (Rombach et al., 2022) to a shape of 4x160x96. The training video is a first-person view of an anonymous player with an in-game ID of "Alex" engaged in the PLAICraft project's multiplayer Minecraft survival world (He, 2024). The evaluation video is a first-person view of another anonymous player with an in-game ID of "Kyrie" who explores

the same Minecraft world. The videos capture multiple continuous play sessions spanning several months within this shared multiplayer survival world, showcasing various biomes in all three world dimensions, mining, crafting activities, construction, mob fighting, and player-to-player interactions. The Minecraft world contains aspects that repeat (ex. day-night cycle, players visiting their homes) and do not repeat (ex. felled trees, player chat logs). Solving *Lifelong PLAICraft* requires learning a **highly nonstationary** environment that changes in **multiple timescales** from a single **correlated** video stream.

## 4. Model and Training Regime

**Model**   We use a U-Net video diffusion model with spatiotemporal attention introduced by Harvey et al. (2022), which individually processes the video frames using residual blocks and captures their temporal dependencies using attention layers. The video diffusion model's U-Net operates on $K$ video frames at a time, where $K = 10$ for *Lifelong Bouncing Balls* and $K = 20$ for the other two datasets. The first $K/2$ video frames are model inputs and the next $K/2$ video frames are prediction targets.

During training, given the first $K/2$ uncorrupted video frames and the second $K/2$ video frames corrupted with Gaussian noise, the model regresses the values of the Gaussian noise (Ho et al., 2020). Specifically, let $x$ be a size $K$ window of consecutive video frames, and $F_\theta$ be the U-Net. The denoising loss function is defined as

$$\ell(\boldsymbol{\theta}, \boldsymbol{x}) = \mathbb{E}_{\boldsymbol{\epsilon}, s} \left\| \boldsymbol{\epsilon} - F_{\boldsymbol{\theta}}(\boldsymbol{x}^{\text{obs}}, \boldsymbol{x}_s^{\text{lat}}, s) \right\|^2, \qquad (1)$$

where $\epsilon \sim \mathcal{N}(0, I)$ is a unit normal random vector, $s \sim \mathcal{U}(1, S)$ is the diffusion noising timestep. The superscripts obs and lat respectively represent the input and predicted part of $\boldsymbol{x}$ i.e., the first and second $K/2$ frames. The subscript $s$ denotes that the video frames are corrupted to the $s$-th diffusion timestep using $\epsilon$ as noise (Ho et al., 2020).

| Method | Train Stream | | | | Test Stream | | | |
|---|---|---|---|---|---|---|---|---|
| | FVD | Loss | minADE | ColorKL | FVD | Loss | minADE | ColorKL |
| Offline Learning | 4.5 ±0.1 | 6.3e-5 ±1e-6 | 1.74 ±0.03 | 0.006 ±0.001 | 4.7 ±0.2 | 6.2e-5 ±2e-6 | 1.71 ±0.06 | 0.006 ±0.001 |
| Lifelong Learning | 4.9 ±0.2 | 6.3e-5 ±2e-6 | 1.82 ±0.03 | 0.005 ±0.0 | 4.7 ±0.2 | 6.2e-5 ±3e-6 | 1.81 ±0.03 | 0.005 ±0.0 |

Table 1. *Lifelong Bouncing Balls (O)* performance metrics computed across two training and three sampling seeds.

| Method | Train Stream | | | | Test Stream | | | |
|---|---|---|---|---|---|---|---|---|
| | FVD | Loss | minADE | ColorKL | FVD | Loss | minADE | ColorKL |
| Offline Learning | 5.8 ±0.3 | 6.5e-5 ±1e-6 | 2.04 ±0.09 | 0.007 ±0.002 | 5.9 ±0.2 | 6.5e-5 ±1e-6 | 2.14 ±0.1 | 0.007 ±0.001 |
| Lifelong Learning | 5.0 ±0.1 | 7.4e-5 ±1e-6 | 2.03 ±0.0 | 0.005 ±0.001 | 5.7 ±0.2 | 7.5e-5 ±1e-6 | 2.06 ±0.0 | 0.005 ±0.0 |

Table 2. *Lifelong Bouncing Balls (C)* performance metrics computed across two training and three sampling seeds.

During sampling, given the first $K/2$ clean video frames and the second $K/2$ video frames filled with Gaussian noise, the model iteratively "denoises" the second $K/2$ frames to produce a plausible continuation of the first $K/2$ frames. This is achieved using Karras et al. (2022)'s stochastic sampler.

For the rest of this section, we represent the entire video stream as $\mathcal{X}$ and the window of $K$ video frames starting from the $i^{\text{th}}$ frame as $\mathcal{X}^{i:i+K}$. In addition, the expectation in Equation (1) is approximated with a single-sample Monte Carlo estimate.

**Baseline: Offline (i.i.d.) Learning**  As a baseline representative of standard video models, we train models in an *Offline Learning* regime. The $t$-th training step loss is

$$\mathcal{L}_t^{\text{offline}}(\boldsymbol{\theta}, \mathcal{X}) = \ell(\boldsymbol{\theta}, \mathcal{X}^{i:i+K}), \quad (2)$$

where $i$ is a uniformly sampled video frame index. The resulting training data resembles i.i.d. segments of $K$ video frames, even though all frames originate from a single autocorrelated stream. When using batch size $N > 1$ (as is common), this i.i.d. sampling is repeated $N$ times.

**Lifelong Learning**  While the Offline Learning regime randomly permutes the data during training, the Lifelong Learning regime presents data to the model *in order*. This setup reflects the desiderata outlined in Section 1, where the model learns from a real-time data stream. The $t$-th training step loss is

$$\mathcal{L}_t^{\text{lifelong}}(\boldsymbol{\theta}, \mathcal{X}) = \ell(\boldsymbol{\theta}, \mathcal{X}^{t:t+K}), \quad (3)$$

i.e., the model receives a sliding window of frames from the single video $\mathcal{X}$. When using batch size $N > 1$, we fill the batch with $N - 1$ copies of the current video $X^{t:t+K}$. This is equivalent to estimating Equation (1) with a larger Monte Carlo budget using multiple samples of $\epsilon$ and $s$, resulting in a reduced-variance estimate of the denoising loss.

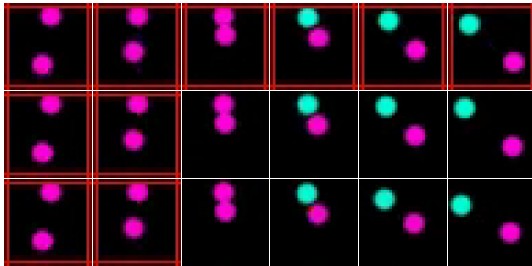

Figure 3. Sample comparison for *Lifelong Bouncing Balls (C)* where ball colors have changed from red/yellow/green. The top, middle, and bottom rows depict videos from the ground truth data, offline learned model, and lifelong learned model respectively.

This Lifelong Learning setup is compatible with numerous techniques proposed by the continual learning community. As a demonstration, we augment the online stream with a replay buffer (Chaudhry et al., 2019) that retains a buffer $\mathcal{M}$ of past video subsequences chosen through reservoir sampling (Vitter, 1985), where the buffer size is chosen to be a small fraction of the total data stream size. With a batch size of $N$, the new $t$-th training step loss is

$$\mathcal{L}_t^{\text{lifelong}}(\boldsymbol{\theta}, \mathcal{X}) = \frac{1}{N}\left( \ell(\boldsymbol{\theta}, \mathcal{X}^{t:t+K}) + \sum_{i=1}^{N-1} \ell(\boldsymbol{\theta}, \mathcal{M}^i) \right), \quad (4)$$

where $\mathcal{M}^i$ is a randomly chosen video from the buffer. One can optionally prioritize the learning of the latest video by adjusting the ratio between the number of replay samples $\mathcal{M}^i$ and copies of $\mathcal{X}^{t:t+k}$ in Equation (4)'s minibatch. Lastly, we emphasize that replay is a simple technique that could be replaced with any other continual learning method.

# 5. Experiments

This section qualitatively and quantitatively compares offline learned and lifelong learned video diffusion models. The learning algorithms use the same batch size and number of gradient steps, equalizing the amount of computation and

| Method | Train Stream | | | Test Stream | | |
|---|---|---|---|---|---|---|
| | FVD | KVD | Loss | FVD | KVD | Loss |
| Offline Learning | 32.2 ±1.1 | 9.2 ±0.8 | 0.0055 ±0.0 | 29.4 ±0.7 | 6.4 ±0.4 | 0.0056 ±0.0 |
| Lifelong Learning | 40.2 ±0.5 | 15.2 ±0.6 | 0.0061 ±0.0001 | 31.7 ±0.7 | 6.3 ±0.4 | 0.0057 ±0.0 |

Table 3. *Lifelong 3D Maze* performance metrics computed across two training and three sampling seeds.

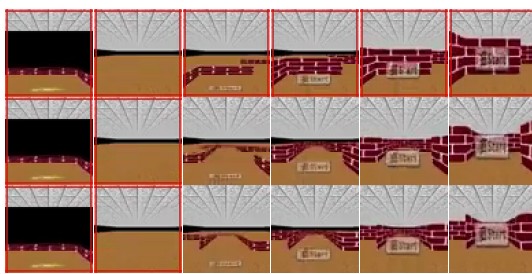

Figure 4. Sample comparison for *Lifelong 3D Maze*'s rising walls rare event. The top, middle, and bottom rows depict videos from the ground truth test stream data, offline learned model, and lifelong learned model respectively.

runtime memory. We refer the readers to Appendix B for additional experiment details.

## 5.1. Lifelong Bouncing Balls.

As discussed in Section 3, the Bouncing Balls dataset features repetitive and deterministic dynamics, with stationary (*Lifelong Bouncing Balls (O)*) and non-stationary (*Lifelong Bouncing Balls (C)*) color transitions.

**Setup**   The models for *Lifelong Bouncing Balls (O)* and *Lifelong Bouncing Balls (C)* respectively have 8 and 74 million parameters. The models are trained with a batch size of 2 and are evaluated using 45 frames that they autoregressively generate conditioned on 5 ground truth frames. Lifelong Learning retains 5 percent of the stream's frames in the replay buffer. We measure the sampled videos' perceptual and temporal coherence using FVD (Unterthiner et al., 2019) and report the loss to measure learning progress. We also measure the models' understanding of the ball movement using minADE (Rasouli, 2020) and of the color transition using our ColorKL metric (Appendix A), extracting the ball positions and colors from the frames using 2D convolutions.

**Result**   Qualitative results appear in Figure 1a and Figure 3. For *Lifelong Bouncing Balls (O)*, Offline Learning and Lifelong Learning produce models that generate visually compelling ball colors and trajectories, correctly handling different ball-to-ball and ball-to-wall collisions. While no model perfectly recovers the deterministic ground truth trajectories, the videos generated by offline and lifelong

learned models are not perceptibly different in quality.[2] Because the model's context size $K = 10$ is too small to always condition on two past ball bounces, the models do not perfectly capture the color transition from red to yellow and green. However, the color transition from yellow to red and green to red are captured well, and the models probabilistically transition from red to yellow or green – the best possible modeling choice given the limited context size. Perhaps surprisingly, we observe the same behaviors for *Lifelong Bouncing Balls (C)*. Despite the non-stationary color changes, both learning algorithms generate realistic ball trajectories and colors, regardless of whether we test on frames from the beginning or the end of the video stream.[3]

The quantitative results in Table 1 and Table 2 mirror the qualitative results. Offline Learning and Lifelong Learning perform similarly on both the stationary *Lifelong Bouncing Balls (O)* and non-stationary *Lifelong Bouncing Balls (C)* datasets. Overall, the presence of temporal correlation and the degree of non-repetitiveness does not pose a significant learning challenge to Lifelong Learning when the video stream is relatively simple.

## 5.2. Lifelong 3D Maze.

The *Lifelong 3D Maze* dataset—a first-person video stream from a partially observable navigation environment—is a step up in complexity from *Lifelong Bouncing Balls*. The frames in the video stream are renderings of a 3D space, whereas in *Lifelong Bouncing Balls* frames consist of 2D objects against a black background. Moreover, as discussed in Section 3, the 3D Maze data stream features randomly generated surroundings as well as rare events.

**Setup**   The diffusion models trained in this section have 78 million parameters. The models are trained with a batch size of 4, of which two elements are the current timestep video frame subsequence, and are evaluated using 40 frames generated from conditioning on 10 ground truth frames. Lifelong Learning retains 5 percent of the video stream's frames in the replay buffer. We measure perceptual and temporal coherence using FVD and KVD (Bińkowski et al., 2021) using the I3D network (Carreira & Zisserman, 2017). We also report the model loss to measure learning progress.

---

[2]*Lifelong Bouncing Balls (O)* model samples are viewable here.
[3]*Lifelong Bouncing Balls (C)* model samples are viewable here.

|  | Train Stream | | | Test Stream | | |
|---|---|---|---|---|---|---|
| Method | FVD | KVD | Loss | FVD | KVD | Loss |
| Offline Learning | 112.6 ±1.1 | 10.4 ±0.3 | 0.0330 | 165.3 ±0.5 | 12.5 ±0.0 | 0.0397 |
| Lifelong Learning | 108.1 ±0.8 | 8.7 ±0.3 | 0.0335 | 167.0 ±1.1 | 13.1 ±0.2 | 0.0411 |

*Table 4. Lifelong PLAICraft* performance metrics computed across one training and three sampling seeds.

**Result**   Qualitative results appear in Figure 1b and Figure 4. Both Offline Learning and Lifelong Learning models generate coherent maze trajectories, successfully modeling sparsely occurring event sequences such as camera inversion and the rising of the maze walls.[4] Both models make the occasional mistake, such as deforming polyhedral gray rock during camera inversion, though neither model makes mistakes more frequently than the other.

Quantitative results appear in Table 3. We find that the Offline and Lifelong Learning performance metrics are generally similar, and that the similarity is more pronounced for the test stream than the train stream. We show in Table 8 that Lifelong Learning's train stream quantitative metrics become nearly identical to Offline Learning's quantitative metrics when we increase the replay buffer size to 20 percent of the data stream. This is because of a subtle perceptual change in the maze texture that occurs within the train stream (see Appendix C), which induces a mild forgetting for the lifelong learned model that disappears with increased replay buffer size. Nevertheless, we see on both qualitative and quantitative fronts that lifelong learned models can capture information on rare events when learning from correlated video frames from a stochastically generated environment, despite heavy data imbalance between rare events and regular maze traversal frames.

### 5.3. Lifelong PLAICraft.

The *Lifelong PLAICraft* dataset is the most challenging of the three datasets. Like *Lifelong 3D Maze*, it is a continual video stream from a partially observable environment. However, the world is more complex with dynamics resulting from interactions between multiple agents.

**Setup**   The diffusion models trained in this section have 80 million parameters. Lifelong Learning retains 20 percent of the video stream's frames in the replay buffer. The models are trained with a batch size of 8, of which two elements are the current timestep video frame subsequence, and are evaluated using 10 frames that they generate conditioned on 10 ground truth frames. We measure the sampled videos' perceptual and temporal coherence using FVD and KVD, and report the model loss to measure learning progress.

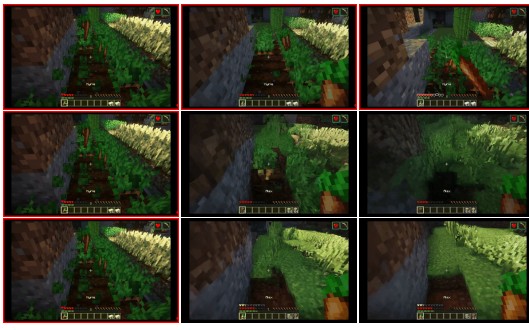

*Figure 5.* Sample comparison for *Lifelong PLAICraft*'s carrot harvesting sequence. The top, middle, and bottom rows depict videos from the ground truth test stream data, offline learned model, and lifelong learned model respectively.

**Result**   See Figure 1c and Figure 5 for qualitative results. Both Offline Learning and Lifelong Learning models capture perceptual details about Minecraft video frames despite having only 80 million parameters and 50 hours of gameplay data to work with. Objects present in every gameplay frame (player name, item bar, and the equipped item) are consistently included in all Offline and Lifelong generations.[5] Both learning paradigms are less successful on this dataset at capturing temporal correlations between frames. We hypothesize that a larger model size and careful hyperparameter tuning achievable on commercial-scale compute will bring the sample quality closer to that of the *Lifelong Bouncing Balls* and *Lifelong 3D Mazes* models. Nevertheless, the generated video frames from Offline and Lifelong models are not perceptually distinguishable in quality.

Quantitative results appear in Table 4. On both training and test streams, Offline Learning and Lifelong Learning perform similarly. To summarize, we find that Lifelong Learning can perform comparably to Offline Learning with minimal hyperparameter tuning, even on data streams as life-like and complex as *Lifelong PLAICraft*.

### 5.4. Discussion

The reported metrics for Offline and Lifelong Learning across all datasets are remarkably similar. To further substantiate this observation, Table 10 in Appendix E presents

---

[4] *Lifelong 3D Maze* model samples are viewable here.

[5] *Lifelong PLAICraft* model samples are viewable here.

two-sided T-test results assessing whether the test stream performance differences between the two learning algorithms are statistically significant. Our analysis reveals that the performance of lifelong-learned video diffusion models is not significantly different from that of offline-learned models. Specifically, for the majority of dataset and metric pairs, the two-sided T-tests fail to reject the null hypothesis, suggesting no strong evidence of a meaningful difference.

Given the closeness in performance of replay-based lifelong learning to i.i.d. video diffusion model learning, it is natural to ask whether this result is sensitive to free parameters in experience replay—the most significant of which is the replay buffer size. We investigate experience replay's sensitivity to the replay buffer size in Appendix D. We find that storing 5 to 20 percent of video stream frames is often sufficient and that unlimited replay buffer size does not lead to significantly stronger results for our diverse group of datasets. We also find that not having a replay buffer impedes model performance and that this behavior is more pronounced when the training stream is nonstationary. These results suggest that experience replay with a modest buffer size is a strong and simple baseline when lifelong learning video diffusion models in memory-constrained settings.

Lastly, to further emphasize the challenging nature of our datasets, we plot the change in the video diffusion models' test stream quantitative metrics during training for all datasets in Appendix F. Both Offline and Lifelong Learned models continue to improve in performance for every metric as training progresses, indicating that the generative modeling tasks cannot easily be mastered after the diffusion models train on a moderate number of video frames.

## 6. Related Work

Our work is closely related to continual generative modeling (Nguyen et al., 2018; Ramapuram et al., 2020; Masip et al., 2024; Smith et al., 2024; Zając et al., 2023; Campo et al., 2020; Chen et al., 2022), in particular to methods that focus on diffusion models and video generative models. Prior work on lifelong learning of diffusion models (Masip et al., 2024; Smith et al., 2024; Zając et al., 2023) focuses on image modeling under the task-based continual learning setup where data grouped into disjoint tasks arrives in large batches. In contrast, our work focuses on lifelong learning of video diffusion models capable of capturing temporal correlations in video frames by learning from a single video stream. Prior work on lifelong learning of video generative models (Campo et al., 2020; Chen et al., 2022) continually learns VAE-based models that can generate future frames again in the task-based continual learning setup. Unlike our setup, these methods assume the presence of rigid task boundaries and the ability to train to convergence on large data batches. In contrast, our lifelong learning datasets do not have notions of tasks, and our models only have access to data as they appear in the stream.

Our work is also related to online learning of sequence processing models (Zucchet et al., 2023; Carreira et al., 2024; Bornschein et al., 2024; Liu et al., 2024). Notably, Carreira et al. (2024) learns predictive video models from the concatenation of loosely related videos. In contrast, our work learns generative video models from a single autocorrelated stream. Liu et al. (2024) learns a linear regression-based world model in an online fashion, an approach that we note is not amenable to high-dimensional videos.

Lastly, work has, like us, introduced video datasets for continual learning. Villa et al. (2022); Tang et al. (2024)'s datasets contain many short videos that can be learned under a task-based continual learning setup. Carreira et al. (2024)'s datasets construct one very long data stream by concatenating multiple short-to-medium length videos, but their data are not publicly available. Singh et al. (2016)'s dataset, while not originally developed for continual learning, contains a series of short-to-medium-length videos of the real world that were collected via Google Glass. We note that their dataset has semantic discontinuities, whereas our datasets come from single, continuous video streams.

## 7. Conclusion

This paper explores lifelong learning of diffusion models from video streams, which has never been investigated to the best of our knowledge. We establish the feasibility of learning video diffusion models from a single autocorrelated video stream in a lifelong fashion. Our lifelong learning approach is simple and uses no specialty techniques beyond a minimal replay buffer. To promote further research into this area, we introduce three datasets that test how different video stream characteristics affect model and lifelong learning algorithm performance.

The ability to effectively train video diffusion models from a continuous data stream holds significant promise, particularly for world model learning (Alonso et al., 2024; Du et al., 2024; Huang et al., 2023; Escontrela et al., 2024) and compute-efficient foundation model adaptation (Smith et al., 2024). Our findings suggest that learning generative models from strongly correlated data streams may not be as difficult as previously assumed, at least for certain data streams and model classes, echoing results from Bornschein et al. (2024) and Carreira et al. (2024) on discriminative model lifelong learning. We expect sample quality to improve significantly with the scale of data and compute, more advanced model architectures, optimization techniques, and continual learning algorithms.

## Impact Statement

This work is relevant to the development of embodied AI agents that can adapt to their environments in real time through lifelong video world model updates. This in principle can greatly enhance the flexibility of the embodied AI agents compared to those that are purely offline trained, which in turn will impact how people perceive and interact with them.

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

## A. The ColorKL Metric

The ColorKL metric measures how faithful a model-generated video's color transition statistics are to the ground truth color transition statistics at the dataset level. It is defined as

$$\frac{1}{|\mathcal{C}|} \sum_{c_{old} \in \mathcal{C}} D_{\mathrm{KL}}\left(p_*(c_{new}|c_{old}) \,\|\, p_{model}(c_{new}|c_{old})\right) \tag{5}$$

where $D_{\mathrm{KL}}$ is the KL divergence, $\mathcal{C} = \{red, yellow, green\}$ is the set of possible ball colors, $p_*(c_{new}|c_{old})$ is the ground truth probability for a ball to switch colors from $c_{old}$ to $c_{new}$, and $p_{model}$ is the empirical probability of color transitions from $c_{old}$ to $c_{new}$ in the model-generated video dataset. We note that if the model's context window is long enough to always capture two past ball bounces for all balls, the ball colors can be deterministically predicted. If the model's context is not long enough, as is the case in our experiments, the ground truth transition probabilities $p_*$ state that the balls always transition from yellow to red and green to red but have a 50/50 chance of transitioning from red to yellow or green.

## B. Additional Experiment Details

The minimum average displacement error (minADE) metric is computed by selecting the trajectory with the lowest average displacement error from 3 sampled trajectories for each evaluation video subsequence. The ColorKL metric is computed by tallying the transition statistics from 3 sampled trajectories for each evaluation video subsequence and computing the KL divergence of this empirical distribution with the ground truth transition statistics. The loss metric is calculated by uniformly sampling 10 noise levels for all evaluation set samples per model checkpoint, computing the diffusion MSE loss, and averaging the results.

As computing all metrics on the entirety of the video streams is prohibitively expensive, we select 1,000 video subsequences from the evaluation stream and calculate the metrics on those video frames for all reported metrics. All training and sampling seeds compute the metrics on the same set of video subsequences. For datasets without significant changes in video frame details throughout the video stream (*Lifelong Bouncing Balls (O)*, *Lifelong 3D Maze*), we select the first 1,000 video subsequences from the evaluation video stream. For datasets with significant changes in video frame details throughout the video stream (*Lifelong Bouncing Balls (C)*, *Lifelong PLAICraft*), we evenly select 1,000 video subsequences across the entire evaluation video stream with equal spacing.

## C. Additional Dataset Curation Details

The train and test set of *Lifelong Bouncing Balls* is generated by randomly sampling the two balls' initial positions and velocities and deterministically updating them to satisfy the conservation of momentum.

The *Lifelong 3D Maze* dataset was created by concatenating two 10-hour-long Lifelong 3D Maze YouTube videos (Dprotp, 2018; Screensavers, 2020) at the point where a maze from the first video is solved and the maze from the second video begins, as the maze screensaver teleports the agent to a newly generated maze on the completion of the previous maze. This concatenation ensures that there is no sudden switch in the environment dynamics (a slight perceptual switch exists as shown in Figure 6).

The *Lifelong PLAICraft* dataset was constructed by concatenating multiple consecutive gameplay session recordings from the same player of the PLAICraft server (He, 2024). All game settings, such as viewing distance, shader configurations, and recording parameters (e.g., resolution and aspect ratio), were consistent across sessions. The dataset includes recordings of two players, Alex and Kyrie, who played in the same survival world alongside hundreds of other players. In their initial gameplay sessions, their starting locations were randomly assigned but could later be altered through travel or teleportation. As in a typical Minecraft survival world, the environment had no boundaries; landscapes were procedurally generated and remained unchanged once created. Player states, including inventories and spawn locations, were preserved between consecutive sessions.

Players' locations and states were generally consistent across most consecutive gameplay sessions. In other words, if a player exited the game at a particular location and state in one session, they would typically resume from the same point in the next session. However, some aspects, such as the player's viewing angle or environment, may be slightly different from reasons such as other players having built structures around the location the player logged off from or other technical reasons related to gameplay recording.

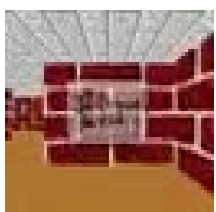

(a) Example frame from video 1.

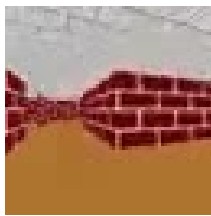

(b) Example frame from video 1.

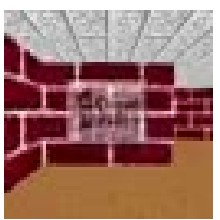

(c) Example frame from video 2.

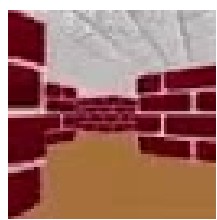

(d) Example frame from video 2.

*Figure 6.* Example video frames from first and second 10-hour-long YouTube videos used to construct *Lifelong 3D Maze*. While the frames are nearly identical, there are very subtle perceptual differences arising from how the uploaders recorded the two videos.

Kyrie's first gameplay session began later than Alex's, and there was some overlap in the locations they visited. Alex generally explored more areas and contributed extensively to building structures, whereas Kyrie primarily played in a village that Alex and other players had previously developed.

## D. Additional Quantitative Results

This section presents the train and test video stream performances of offline training (Offline Learning), lifelong learning using experience replay with limited buffer size (Experience Replay)[6], lifelong learning using experience replay with unlimited buffer size (Full Replay), and lifelong learning without the use of a replay buffer (No Replay) on all datasets.

### D.1. Lifelong Bouncing Balls

| Method | Train Stream | | | | Test Stream | | | |
|---|---|---|---|---|---|---|---|---|
| | FVD | Loss | minADE | ColorKL | FVD | Loss | minADE | ColorKL |
| Offline Learning | 4.5 ±0.1 | 6.3e-5 ±1e-6 | 1.74 ±0.03 | 0.006 ±0.001 | 4.7 ±0.2 | 6.2e-5 ±2e-6 | 1.71 ±0.06 | 0.006 ±0.001 |
| No Replay | 4.7 ±0.1 | 6.7e-5 ±1e-6 | 1.91 ±0.01 | 0.006 ±0.0 | 5.0 ±0.2 | 6.7e-5 ±1e-6 | 1.84 ±0.03 | 0.005 ±0.0 |
| Experience Replay | 4.9 ±0.2 | 6.3e-5 ±2e-6 | 1.82 ±0.03 | 0.005 ±0.0 | 4.7 ±0.2 | 6.2e-5 ±3e-6 | 1.81 ±0.03 | 0.005 ±0.0 |
| Full Replay | 4.7 ±0.1 | 6.3e-5 ±1e-6 | 1.80 ±0.01 | 0.004 ±0.001 | 4.6 ±0.1 | 6.2e-5 ±0.0 | 1.76 ±0.02 | 0.003 ±0.001 |

*Table 5. Lifelong Bouncing Balls (O)* performance metrics. The left and right columns respectively denote training and test video stream results computed across two training and three sampling random seeds.

For *Lifelong Bouncing Balls (O)*, all learning algorithms attain largely indistinguishable results. However, for *Lifelong Bouncing Balls (C)*, No Replay performs significantly worse on all metrics than the rest as it suffered from catastrophic forgetting of ball colors observed earlier in the training stream as shown in Figure 8b. This suggests that when lifelong learning on video streams that contain non-repeating details, mechanisms that preserve past knowledge are necessary regardless of how simple the video streams might be.

### D.2. Lifelong 3D Maze

Table 7 shows how the similarity between offline training and memory-constrained experience replay methods is more pronounced on the test stream than the train stream. This is in part due to the way the *Lifelong 3D Maze* dataset is constructed.

---

[6]This configuration is referred to as Lifelong Learning in the main text's Section 5.

| Method | Train Stream | | | | Test Stream | | | |
|---|---|---|---|---|---|---|---|---|
| | FVD | Loss | minADE | ColorKL | FVD | Loss | minADE | ColorKL |
| Offline Learning | 5.8 ±0.3 | 6.5e-5 ±1e-6 | 2.04 ±0.09 | 0.007 ±0.002 | 5.9 ±0.2 | 6.5e-5 ±1e-6 | 2.14 ±0.1 | 0.007 ±0.001 |
| No Replay | 357.4 ±1.8 | 2.2e-2 ±1e-3 | 2.61 ±0.08 | 0.021 ±0.002 | 343.6 ±1.2 | 2.3e-2 ±1e-3 | 2.73 ±0.11 | 0.022 ±0.0 |
| Experience Replay | 5.0 ±0.1 | 7.4e-5 ±1e-6 | 2.03 ±0.00 | 0.005 ±0.001 | 5.7 ±0.2 | 7.5e-5 ±1e-6 | 2.06 ±0.0 | 0.005 ±0.0 |
| Full Replay | 5.0 ±0.1 | 8.0e-5 ±1e-6 | 2.08 ±0.03 | 0.005 ±0.001 | 5.6 ±0.2 | 7.9e-5 ±2e-6 | 2.12 ±0.0 | 0.006 ±0.001 |

*Table 6. Lifelong Bouncing Balls (C)* performance metrics. The left and right columns respectively denote training and test video stream results computed across two training and three sampling random seeds.

| Method | Train Stream | | | Test Stream | | |
|---|---|---|---|---|---|---|
| | FVD | KVD | Loss | FVD | KVD | Loss |
| Offline Learning | 32.2 ±1.1 | 9.2 ±0.8 | 0.0055 ±0.0 | 29.4 ±0.7 | 6.4 ±0.4 | 0.0056 ±0.0001 |
| No Replay | 130.1 ±1.0 | 84.6 ±1.1 | 0.0085 ±0.0002 | 37.0 ±0.9 | 8.9 ±0.7 | 0.0059 ±0.0 |
| Experience Replay | 40.2 ±0.5 | 15.2 ±0.6 | 0.0061 ±0.0001 | 31.7 ±0.7 | 6.3 ±0.4 | 0.0057 ±0.0 |
| Full Replay | 35.6 ±0.8 | 12.4 ±0.7 | 0.0058 ±0.0001 | 34.8 ±0.3 | 9.2 ±0.2 | 0.0058 ±0.0001 |

*Table 7. Lifelong 3D Maze* performance metrics. Results are computed across two training and three sampling random seeds.

| Replay Buffer Size | Train Stream | | | Test Stream | | |
|---|---|---|---|---|---|---|
| | FVD | KVD | Loss | FVD | KVD | Loss |
| 0 | 130.1 ±1.0 | 84.6 ±1.1 | 0.0085 ±0.0002 | 37.0 ±0.9 | 8.9 ±0.7 | 0.0059 ±0.0 |
| 10,000 | 80.9 ±0.2 | 42.3 ±0.2 | 0.0086 ±0.0 | 39.6 ±0.9 | 11.0 ±0.7 | 0.0060 ±0.0 |
| 50,000 | 40.2 ±0.5 | 15.2 ±0.6 | 0.0061 ±0.0001 | 31.7 ±0.7 | 6.3 ±0.4 | 0.0057 ±0.0 |
| 200,000 | 32.9 ±0.8 | 10.3 ±1.1 | 0.0057 ±0.0 | 32.6 ±0.6 | 8.2 ±0.2 | 0.0057 ±0.0 |
| 1,000,000 | 35.6 ±0.8 | 12.4 ±0.7 | 0.0058 ±0.0001 | 34.8 ±0.3 | 9.2 ±0.2 | 0.0058 ±0.0001 |

*Table 8. Lifelong 3D Maze* performance metrics for experience replay with varying replay buffer sizes.

The dataset is built by concatenating two 10-hour-long YouTube videos, and the test set only comprises unobserved frames from the second video. While the videos are nearly identical, there are very subtle perceptual differences (refer to Figure 6). Because the training stream contains frames from the first and second videos and the test stream contains frames from the second video, lifelong learning methods with limited replay buffer sizes perform better on the test stream videos than on the train stream videos as it overfits to the perceptual details of the second video. This is verified in Table 8, where increasing the replay buffer size is shown to improve train stream metrics much more than the test stream metrics.

### D.3. Lifelong PLAICraft

| Method | Train Stream | | | Test Stream | | |
|---|---|---|---|---|---|---|
| | FVD | KVD | Loss | FVD | KVD | Loss |
| Offline Learning | 112.6 ±1.1 | 10.4 ±0.3 | 0.0329 | 165.3 ±0.5 | 12.5 ±0.0 | 0.0397 |
| No Replay | 512.6 ±3.6 | 39.0 ±0.5 | 0.0513 | 608.1 ±9.5 | 58.7 ±1.2 | 0.0507 |
| Experience Replay | 108.1 ±0.8 | 8.7 ±0.3 | 0.0335 | 167.0 ±1.1 | 13.1 ±0.2 | 0.0411 |
| Full Replay | 112.9 ±1.1 | 10.2 ±0.5 | 0.0331 | 161.3 ±1.3 | 11.3 ±0.5 | 0.0407 |

*Table 9. Lifelong PLAICraft* performance metrics. Results are computed across one training and three sampling seeds.

We observe that offline training and experience replay performances are indistinguishable, but No Replay performs significantly worse. Figure 8d shows the final models' performances on video frames at different training stream indices. To further analyze model behaviors, we plot the fully trained models' performance metrics on video frames from different parts of the training stream. Interestingly, the relative performance difference between different training stream frame indices for the lifelong learned models, even No Replay, mirrors that of the offline-learned model. This suggests that video stream underfitting, not forgetting, is the primary challenge in effective learning of *Lifelong PLAICraft* as we can assume that the offline-learned model does not suffer from forgetting.

## E. Test Stream Statistical Significance Testing

| Dataset | FVD | KVD | Loss | minADE | ColorKL |
|---|---|---|---|---|---|
| *Lifelong Bouncing Balls (O)* | 0.932 | - | 0.745 | 0.310 | 0.663 |
| *Lifelong Bouncing Balls (C)* | 0.456 | - | **0.000** | 0.563 | 0.101 |
| *Lifelong 3D Maze* | 0.039 | 0.896 | 0.020 | - | - |
| *Lifelong PLAICraft* | 0.270 | 0.145 | 0.073 | - | - |

*Table 10.* p-values from two-sided T-test where the null hypothesis that there is no difference between the test stream performance metrics from Offline Learning and Lifelong Learning. Dataset and metric pairs where the null hypothesis is rejected with the significant level $\alpha = 0.05$ are underlined. Dataset and metric pairs where the null hypothesis is rejected with the significant level $\alpha = 0.02$ are bolded.

Table 10 details two-sided T-test results that compare whether the quantitative performance differences between Offline Learning and Lifelong Learning are statistically significant for all dataset and metric pairs. Adhering to the main experiment setup, the *Lifelong Bouncing Balls* and *Lifelong 3D Maze* results were computed from samples generated from two training and three sampling seeds, and the *Lifelong PLAICraft* results were computed from samples generated from one training and three sampling seeds. We find that, for most cases, the two-sided T-tests fail to reject the null hypothesis that the performance difference between the two learning algorithms is statistically insignificant.

# F. Training Time Performance Metrics

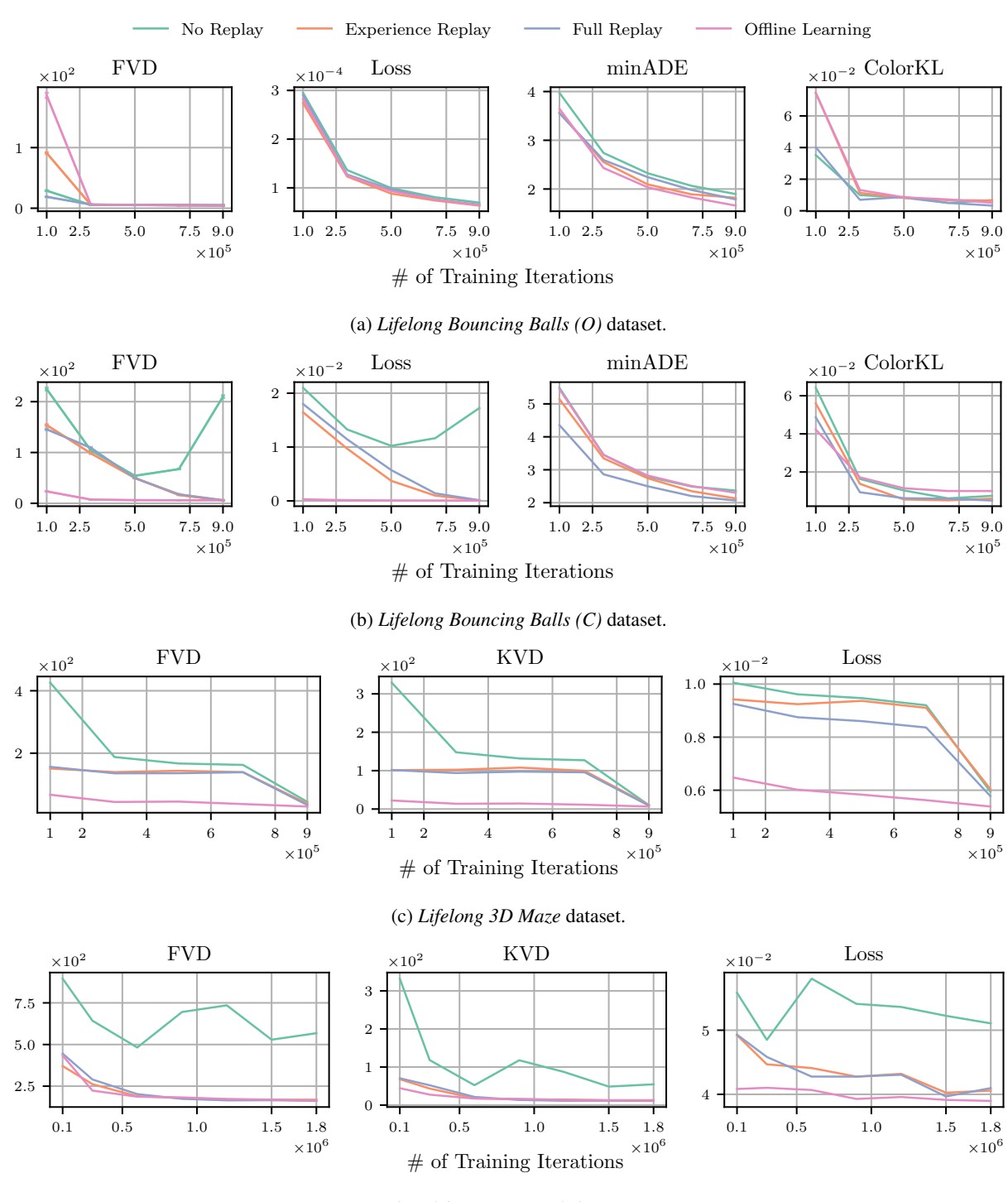

(a) *Lifelong Bouncing Balls (O)* dataset.

(b) *Lifelong Bouncing Balls (C)* dataset.

(c) *Lifelong 3D Maze* dataset.

(d) *Lifelong PLAICraft* dataset.

*Figure 7.* Test stream performance metrics for model checkpoints at different training iterations. The plots show the improvement in model quality as online training progresses. All models generally improve the longer they are trained.

# G. Train Stream Performance Breakdown

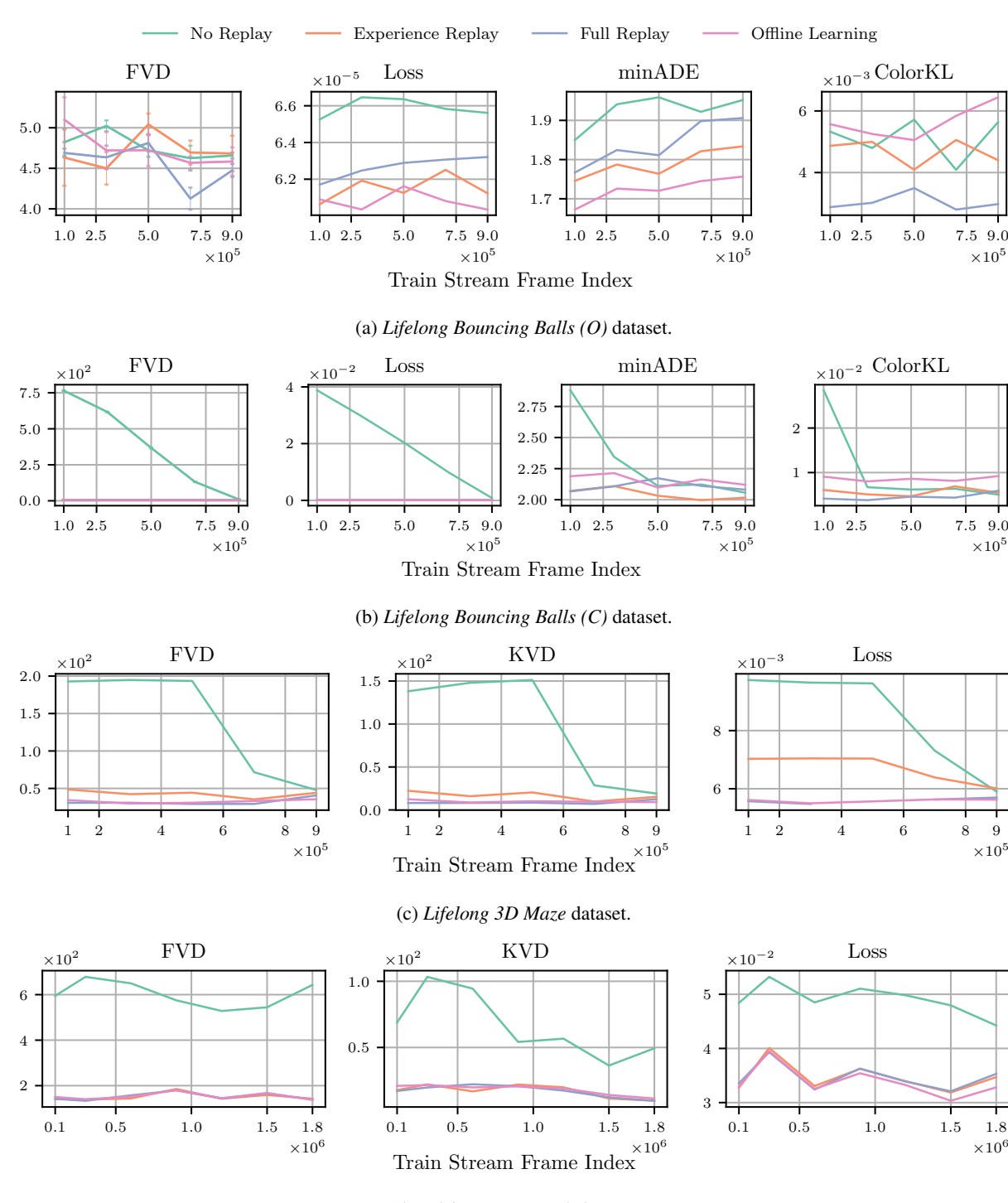

(a) *Lifelong Bouncing Balls (O)* dataset.

(b) *Lifelong Bouncing Balls (C)* dataset.

(c) *Lifelong 3D Maze* dataset.

(d) *Lifelong PLAICraft* dataset.

*Figure 8.* Fully trained models' performance metrics on video frames from different parts of the training stream. The plots show whether the final model performs better or worse on future frame prediction for frames earlier or later in the training stream. Each point of the plots is calculated using 1,000 consecutive video frame subsequences that succeed the corresponding train stream frame index.

