# OpenReview forum: "Lifelong Learning of Video Diffusion Models From a Single Video Stream"
_ICML.cc/2025/Conference — Submitted to ICML 2025_

### Official Review · Reviewer_YRxK · 2025-03-10

**Overall Recommendation:** 3

**Summary:**

This work proposes learning a video diffusion model from a single video stream using lifelong learning, specifically through experience replay. The results demonstrate improved performance compared to standard diffusion training. Additionally, three benchmarks are introduced to support the experiments.

**Claims And Evidence:**

Yes, the experimental results strongly support the proposed lifelong learning claim.

**Essential References Not Discussed:**

I have concerns regarding the relevance of this work to the video prediction and world model literature.

**Experimental Designs Or Analyses:**

The experimental results are robust across the three proposed datasets.

**Methods And Evaluation Criteria:**

The method is simple and aligns well with lifelong learning. However, the proposed datasets seem somewhat limited, as the scenarios covered are relatively narrow.

**Other Comments Or Suggestions:**

No.

**Other Strengths And Weaknesses:**

Strengths:
1. The paper introduces a lifelong learning paradigm for training video diffusion models.
2. It proposes three datasets specifically designed to address the problem.

Weaknesses:
1. Limited Comparisons: This work is highly related to world models such as Genex, GameFactory, GameGen-X, and other world model-based video generation approaches. Additionally, video prediction is another closely related field. However, the paper does not provide comparisons with these works, despite the fact that the proposed datasets share strong similarities with them.
2. Necessity of This Approach: In foundational video generation training, the primary challenge is the lack of a single long continuous video stream, as datasets typically contain frequent scene changes, resulting in only a few-second clips. While continual learning and lifelong learning are important topics, their relevance to video generation seems more crucial in game generation settings, such as in Genie. The necessity of applying lifelong learning to general video generation remains unclear.

**Questions For Authors:**

Please include video samples in the supplementary material rather than providing Google Drive links.

**Relation To Broader Scientific Literature:**

This work is the first to propose training a video diffusion model in a lifelong learning framework within the community. However, I question the necessity of this approach.

**Theoretical Claims:**

This work does not propose any theoretical claims.

---

> ### Author Rebuttal · Authors · 2025-03-31
>
> Before addressing specific concerns, we note that several reviewers have entirely missed the Google Drive links [[a](https://drive.google.com/drive/folders/1ToqSvdFsXJm0UqJZlRURI1uwsIvbHYHn),[b](https://drive.google.com/drive/folders/1IopUyb98v0ybqlaCtimayc9RXnMG63Y-),[c](https://drive.google.com/drive/folders/1Lz54gCnavXmsoFYv9EQEcd0n4mIt91Gi),[d](https://drive.google.com/drive/folders/19CQQShbv3dm04kVk5n4Xf8DvfmJo-9uR)] (Section 5 Footnotes 2,3,4,5) containing video samples from Lifelong and Offline Learning models for every dataset (lower quality videos viewable online [here](https://drive.google.com/drive/folders/1nlc1NZG8lFZIE7w0hYsa2Np-YXjKjSJl)). Some reviewers also missed the reference to the quantitative results for two more baselines in the appendix for every dataset (Section 5 Line 398, Appendix D). These oversights have led to lower scores and skepticism regarding our main claim—that lifelong learning of video diffusion models is not only feasible but can achieve performance comparable to offline training given the same number of gradient steps. We kindly ask that you visit these resources should you question the validity of our claim and re-evaluate your scores in light of these results.
>
> To substantiate our main claim even further, we have incorporated new results on a real-world driving dataset despite operating within the limits of resources available to an academic lab. Please refer to the rebuttal addressed to Reviewer KLJW for the experiment setup and qualitative and quantitative results. These results support the validity of our claim on complex real-world video data.
>
> We now address specific questions and concerns.
>
> > The paper does not provide comparisons with world models and video prediction models, despite the fact that the proposed datasets share strong similarities with them.
>
> We agree that our lifelong learning of autoregressive video generative models is related to those two problems. However, it is difficult to directly compare our lifelong models to the baselines from those problems. World models require action conditioning, and the introduced video datasets do not contain actions. Video prediction models are incapable of generating multiple data samples, so it's not straightforward to compare against them via video generative modeling metrics. However, given the relevance to our problem setup, we will cite these world model papers in the camera-ready draft.
> > While continual learning and lifelong learning are important topics, the relevance of the proposed single long, continuous video streams seems more crucial in game generation settings, such as in Genie. The necessity of applying lifelong learning to general video generation remains unclear.
>
> We are glad that the reviewer finds single long, continuous video streams valuable to game generation settings and highlight that the proposed datasets (ex. Lifelong PLAICraft) could also benefit the game generation community. As the reviewer noted earlier, autoregressive video modeling is intimately related to world modeling since we can view the former as the latter marginalized over actions. The key motivation behind this work is to make progress toward lifelong learnable vision-based embodied AI whose world model can be updated in real-time. We believe that agents with updatable world models that are combined with planners, for example a language model as in GenEx, is key to having flexible embodied AI agents that are not constrained to the behaviors associated with pretraining checkpoints or can only be updated periodically. We chose to work with video modality in the beginning because it is much easier to generate datasets that work purely on video modalities, and we expect action conditioning to not significantly perturb our results.
> > Please include video samples in the supplementary material rather than providing Google Drive links.
>
> Thank you for your comment. We will update this for the camera-ready draft.
>
> ---
>
> We appreciate that the reviewer believes that our experimental results are robust and that they strongly support our claim. We hope that our rebuttal experimental results, which further support the fact that lifelong learning of video diffusion models is possible for complex, real-life datasets, addresses the reviewer’s concern that the previously covered scenarios are relatively narrow. In addition, we hope that our rebuttal’s clarification on the motivation of our work being the first step toward lifelong learnable vision-based embodied AI clarifies the value of our work. Thank you again for your engagement and we would be delighted to have further discussions and address any other questions.

---

### Official Review · Reviewer_KLJW · 2025-03-13

**Overall Recommendation:** 1

**Summary:**

This work shows that autoregressive video diffusion models can be effectively trained from a single continuous video stream, matching the performance of standard offline training given the same number of gradient steps. The authors further demonstrate that this result can be achieved using experience replay with a limited subset of past frames. Additionally, we introduce three new datasets designed for evaluating lifelong video model learning: Lifelong Bouncing Balls, Lifelong 3D Maze, and Lifelong PLAICraft. Each dataset consists of over a million consecutive frames, capturing environments of increasing complexity.

**Claims And Evidence:**

The paper's central claim—that training video diffusion models in a lifelong manner from a single continuous video stream is as effective as offline training—is not sufficiently supported by robust evidence. The experiments rely on a narrow set of synthetic datasets, which fail to capture the complexity and variability of real-world video streams, making the generalizability of the approach highly questionable. Furthermore, the evaluation lacks critical ablations on forgetting, stability over long horizons, and model degradation, leaving the key claims speculative rather than convincingly demonstrated.

**Essential References Not Discussed:**

Not found

**Experimental Designs Or Analyses:**

The evaluation metrics (FVD, minADE, ColorKL) are standard for assessing generative video models, but additional robustness tests on long-term temporal consistency and adaptation to domain shifts would enhance credibility. While the synthetic datasets capture key challenges like non-stationarity and temporal correlation, real-world video streams with more diverse dynamics would better reflect practical deployment scenarios.

**Methods And Evaluation Criteria:**

The proposed method—experience replay with limited memory—aligns well with the goal of lifelong video learning, as it helps mitigate catastrophic forgetting while maintaining efficiency.

**Other Comments Or Suggestions:**

No

**Other Strengths And Weaknesses:**

Strength:

1. The paper investigates lifelong learning for video diffusion models, a relatively unexplored area within generative modeling. While the approach is simple, demonstrating that lifelong learning can approximate offline training is an interesting empirical finding that may inspire future research in more complex settings.

2. The paper presents a well-structured experimental setup, including a fair comparison between lifelong and offline training using the same number of gradient steps.

Weakness:

1. The proposed method primarily relies on experience replay, a well-established technique in continual learning, without introducing significant modifications or improvements. The lack of new architectural contributions or theoretical insights reduces the paper’s originality and impact on the field.

2. Synthetic Datasets Reduce Practical Relevance: the evaluation relies solely on synthetic datasets, which, while structured, fail to reflect the complexity, variability, and challenges of real-world video streams. Without testing on more diverse and dynamic environments, it remains unclear whether the proposed lifelong learning approach generalizes beyond controlled settings.

3. Overclaim: The paper claims that lifelong learning performs comparably to offline training, but does not provide sufficient ablations on long-term stability, forgetting, or sensitivity to different training conditions. The absence of experiments on how the model adapts to distribution shifts or scales with increasing data exposure weakens the validity of its claims.

**Questions For Authors:**

NA

**Relation To Broader Scientific Literature:**

While it provides an empirical proof-of-concept that lifelong training can approximate offline training, its reliance on experience replay without deeper investigation into forgetting, stability, or adaptation makes it a limited contribution.

**Theoretical Claims:**

No theoretical claims are required for review in this work.

---

> ### Author Rebuttal · Authors · 2025-03-31
>
> Before addressing specific concerns, we note that several reviewers have entirely missed the Google Drive links [[a](https://drive.google.com/drive/folders/1ToqSvdFsXJm0UqJZlRURI1uwsIvbHYHn),[b](https://drive.google.com/drive/folders/1IopUyb98v0ybqlaCtimayc9RXnMG63Y-),[c](https://drive.google.com/drive/folders/1Lz54gCnavXmsoFYv9EQEcd0n4mIt91Gi),[d](https://drive.google.com/drive/folders/19CQQShbv3dm04kVk5n4Xf8DvfmJo-9uR)] (Section 5 Footnotes 2,3,4,5) containing video samples from Lifelong and Offline Learning models for every dataset (lower quality videos viewable online [here](https://drive.google.com/drive/folders/1nlc1NZG8lFZIE7w0hYsa2Np-YXjKjSJl)). Some reviewers also missed the reference to the quantitative results for two more baselines in the appendix for every dataset (Section 5 Line 398, Appendix D). These oversights have led to lower scores and skepticism regarding our main claim—that lifelong learning of video diffusion models is not only feasible but can achieve performance comparable to offline training given the same number of gradient steps. We kindly ask that you visit these resources should you question the validity of our claim and re-evaluate your scores in light of these results.
>
> To substantiate our main claim even further, we have incorporated new results on a real-world driving dataset despite operating within the limits of resources available to an academic lab. Specifically, we evaluated Offline and Lifelong Learning on a continuous video stream comprising 550K training and 20K test frames, recorded from a single car’s dashcam over multiple driving sessions totalling 8 hours. We refer to this dataset as **Lifelong Drive** and will release it upon acceptance. The dataset consists of 1 to 40-minute driving sessions, with transitions occurring when the car is started and parked. Fade-in and out animations are applied at the video concatenation boundaries to ensure smooth session transitions. Frames are encoded into 4×64×64 latents using the Stable Diffusion encoder. Other experiment details are identical to the Lifelong PLAICraft experiment.
>
> Lifelong Drive qualitative results are presented [here](https://drive.google.com/drive/folders/1BP6Uqr8R7979ZvbEU9JTGkW83v_lr2jq), and the quantitative results are below. The samples produced by Offline and Lifelong Learning are qualitatively indistinguishable and quantitatively comparable.
> |Method|Train FVD|Train KVD|Train Loss|Test FVD|Test KVD|Test Loss|
> |-|-|-|-|-|-|-|
> |Offline Learning|25.9 ± 0.6|2.7 ± 0.1|0.0299 ± 0.0002|36.2 ± 2.3|12.8 ± 3.2|0.0311 ± 0.0002|
> |Lifelong Learning|23.0 ± 0.3|0.9 ± 0.2|0.0303 ± 0.0004|33.7 ± 1.2|9.8 ± 1.7|0.0316 ± 0.0003|
>
> In the style of Appendix E, we also report p-values from two-sided T-tests where the null hypothesis is that there is no difference between the test stream performance metrics from Offline and Lifelong Learning. The tests fail to reject the null hypothesis that the performances of the two algorithms are not different ($\alpha=0.05$).
> |Dataset|FVD|KVD|Loss|
> |-|-|-|-|
> |Lifelong Drive|0.407|0.466|0.219|
>
> We now address specific questions and concerns.
> > There are no improvements to experience replay nor ablations on forgetting.
>
> Please refer to the last and first points of our rebuttal addressed to Reviewer gPNY.
> > There are no ablations on long horizon stability nor sensitivity to different training conditions.
>
> We present new qualitative results that show the similarity in the quality of Lifelong and Offline learning's long video samples [here](https://drive.google.com/drive/folders/1KlHgtNsP_p1q6-rzJ_Dq-acXqyWa0Bkm).
> Our original results also show that Lifelong and Offline Learning perform comparably across datasets, highlighting the former's robustness to training conditions.
> > The absence of experiments on how the model adapts to distribution shifts or scales with increasing data exposure weakens the validity of its claims.
>
> Distribution shift is synonymous with nonstationarity. Lifelong Bouncing Balls (O), (C) results in Appendix D localizes the effect of color nonstationarity on all baselines, while Lifelong PLAICraft measures the effect of distribution shift in multiple timescales. Appendix G shows the forgetting behaviors associated with distribution shift. Lastly, Appendix F illustrates how model performance on test stream scales with increasing data exposure for all datasets.
> > FVD, minADE, ColorKL are standard for assessing generative video models.
>
> We note that minADE (trajectory metric) and ColorKL (our original metric) as video metrics are not standard.
>
> ---
>
> We appreciate that the reviewer finds our experiments well-structured and our findings interesting. We hope that our rebuttal results and analysis from the appendix will help the reviewer share Reviewer YRxK’s sentiment that our results strongly support the claim that training autoregressive video diffusion models from a single video stream can be as effective as offline training given the same number of gradient steps.

---

### Official Review · Reviewer_gPNY · 2025-03-14

**Overall Recommendation:** 1

**Summary:**

This work investigates ability to learn a video diffusion model in non-iid setting - from a single continuous video stream.
The overall method is an autoregressive UNet-based video diffusion model trained with a continuous stream of data and equipped with a replay buffer. Authors also introduce a collection of 3 synthetic datasets to validate their method and show performance on par with offline(iid) training.

**Claims And Evidence:**

This paper claims that learning a video diffusion model on a video stream works. They do demonstrate this on a single architecture and lifelong learning setup (stream+replay buffer), and do not provide any actual generated video outputs - which is not fully convincing.

**Essential References Not Discussed:**

N/A

**Experimental Designs Or Analyses:**

Given that there is no theoretical analysis of any sort, or any claim on technical novelty, it seems like experimental evaluation is severely lacking - no ablation study is present.

**Methods And Evaluation Criteria:**

Overall method makes sense, the datasets and metrics are also meaningful, but the only baseline is the iid method seems to be insufficient to get an understanding.

**Other Comments Or Suggestions:**

I find it strange that no visual results are provided in a supplementary.

**Other Strengths And Weaknesses:**

+ The key result of the paper is indeed interesting.
- This work does not claim or introduce any technical novelties, apart from applying an existing technique to a somewhat different context.
- Experimental evaluation does not include any ablations (on architecture / objective / context length) - and thus is not particularly informative.

**Questions For Authors:**

- How critical is a replay buffer?
- Would a choice of the underlying architecture matter?

**Relation To Broader Scientific Literature:**

Method combines a relatively recent VDM with an existing approach for lifelong learning (replay buffer).

**Theoretical Claims:**

N/A

---

> ### Author Rebuttal · Authors · 2025-03-31
>
> Before addressing specific concerns, we note that several reviewers have entirely missed the Google Drive links [[a](https://drive.google.com/drive/folders/1ToqSvdFsXJm0UqJZlRURI1uwsIvbHYHn),[b](https://drive.google.com/drive/folders/1IopUyb98v0ybqlaCtimayc9RXnMG63Y-),[c](https://drive.google.com/drive/folders/1Lz54gCnavXmsoFYv9EQEcd0n4mIt91Gi),[d](https://drive.google.com/drive/folders/19CQQShbv3dm04kVk5n4Xf8DvfmJo-9uR)] (Section 5 Footnotes 2,3,4,5) containing video samples from Lifelong and Offline Learning models for every dataset (lower quality videos viewable online [here](https://drive.google.com/drive/folders/1nlc1NZG8lFZIE7w0hYsa2Np-YXjKjSJl)). Some reviewers also missed the reference to the quantitative results for two more baselines in the appendix for every dataset (Section 5 Line 398, Appendix D). These oversights have led to lower scores and skepticism regarding our main claim—that lifelong learning of video diffusion models is not only feasible but can achieve performance comparable to offline training given the same number of gradient steps. We kindly ask that you visit these resources should you question the validity of our claim and re-evaluate your scores in light of these results.
>
> To substantiate our main claim even further, we have incorporated new results on a real-world driving dataset despite operating within the limits of resources available to an academic lab. Please refer to the rebuttal addressed to Reviewer KLJW for the experiment setup and qualitative and quantitative results. These results support the validity of our claim on complex real-world video data.
>
> We now address specific questions and concerns.
>
> > The only baseline is the iid method and no ablation study is present (related: how critical is the replay buffer?).
>
> We summarize the importance of replay buffer in the main text (Section 5.4, Line 398) where we point the readers to Appendix D where, for all datasets, we report and elaborate on the quantitative results for two additional baselines: naive sliding window lifelong learning that ablates replay loss from the replay objective (No Replay) and unlimited memory experience replay (Full Replay). Additional replay buffer size results for Lifelong 3D Maze are presented in Table 8. Furthermore, Appendix F and G respectively showcase how fast different baselines can learn to perform well on the test stream and how much forgetting affects the baselines’ final models on the train stream for all datasets.
>
> > The paper demonstrates its finding on a single architecture. Would the choice of the underlying architecture affect the results?
>
> Given the related work [1] that demonstrates how both U-Net and Transformer-based non-generative models can be online learned on a data stream made from stitching short, unrelated videos, we expect the architecture to not significantly affect the results. We underscore that we have successfully demonstrated that it is possible for lifelong learned video diffusion models with tens of millions of parameters to achieve a performance good enough to be comparable to offline learning across multiple datasets and two parameter sizes, a finding that was not previously evident.
> > This work does not claim or introduce any technical novelties, apart from applying an existing technique to a somewhat different context.
>
> We believe that the machine learning community will nonetheless benefit from being aware of our novel investigation into a different kind of video generative model training regime for both its carefully designed datasets and its surprising findings. Our streaming learning setup is particularly relevant to the development of lifelong learnable vision-based embodied AI that can update its generative predictive model in real time. In addition, although our focus is on investigating a new problem setup, no prior work has introduced batch-level duplication of the latest sliding window with different noising levels for better estimation of the streaming diffusion loss in Equations (3) to our knowledge.
>
> ---
>
> We appreciate that the reviewer finds the key result of the paper interesting. We hope that our video model samples, additional baselines, analysis, and results will help the reviewer share Reviewer YRxK’s sentiment that our results strongly support the claim that training autoregressive video diffusion models from a single video stream can be as effective as offline training given the same number of gradient steps.
>
> [1] Carreira, J., King, M., Patraucean, V., Gokay, D., Ionescu, C., Yang, Y., Zoran, D., Heyward, J., Doersch, C., Aytar, Y., Damen, D., and Zisserman, A. Learning from one continuous video stream, 2024.

---

### Official Review · Reviewer_gLa1 · 2025-03-16

**Overall Recommendation:** 2

**Summary:**

This study shows that autoregressive video diffusion models can be effectively trained from a single, continuous video stream, matching the performance of standard offline methods given the same number of gradient steps. The key lies in using experience replay that retains only a subset of preceding frames. Additionally, the authors introduce three new lifelong video model learning datasets—Lifelong Bouncing Balls, Lifelong 3D Maze, and Lifelong PLAICraft—each containing over a million consecutive frames of increasing complexity.

**Claims And Evidence:**

I don‘t think claims in the submission are supported by clear and convincing evidence due to unclear and extreme experiment settings. Please see weakness part for details.

**Essential References Not Discussed:**

Missing several important references.

1. Diffusion Forcing: Next-token Prediction Meets Full-Sequence Diffusion. NeurIPS
2. GameGen. ICML
3. Genie.  ICML

**Experimental Designs Or Analyses:**

Yes. I checked the experimental but I don't think is clear and soundess. Please see weakness part for details

**Methods And Evaluation Criteria:**

Yes.

**Other Comments Or Suggestions:**

NA

**Other Strengths And Weaknesses:**

Strengths

The idea is interesting and novel.

The writing is clear and easy to follow.

Weaknesses

It appears that t in Equation (3) for lifelong learning exactly matches the total duration of the training video. In the comparison between lifelong learning and offline training, does i in Equation (2) cover the entire duration of the training video despite being randomly selected? Ensuring this would make the comparison fair.

Since the right-hand side of Equation (4) is also randomized, if the training steps greatly exceed the actual t in the training video (e.g., by a factor of 10), there is no difference between the lifelong loss and the offline loss. How does the comparison between lifelong and offline methods hold under these much longer training steps?

Most open-sourced video models currently exceed 500M parameters. It is unclear whether the same outcomes would apply for larger video U-Nets (beyond 10M or 100M parameters) rather than the smaller ones used in the paper.

Several recent papers (e.g., Diffusion Force, Genie, Genie2, Gamegen) demonstrate that offline training for autoregressive + diffusion (conditioning on past frames) can generate robust, infinite single-game simulations, which are more complex than the datasets presented here. These relevant works are missing from the paper.

There is a lack of visual results, which are essential for a video-focused study.

**Questions For Authors:**

Please see weakness part

**Relation To Broader Scientific Literature:**

If these claims were substantiated by validated experiments, then lifelong learning for training a video diffusion model could make a valuable contribution to the broader scientific literature; however, that unfortunately is not the case.

**Theoretical Claims:**

NO Theoretical Claims.

---

> ### Author Rebuttal · Authors · 2025-03-30
>
> Before addressing specific concerns, we note that several reviewers have entirely missed the Google Drive links [[a](https://drive.google.com/drive/folders/1ToqSvdFsXJm0UqJZlRURI1uwsIvbHYHn),[b](https://drive.google.com/drive/folders/1IopUyb98v0ybqlaCtimayc9RXnMG63Y-),[c](https://drive.google.com/drive/folders/1Lz54gCnavXmsoFYv9EQEcd0n4mIt91Gi),[d](https://drive.google.com/drive/folders/19CQQShbv3dm04kVk5n4Xf8DvfmJo-9uR)] (Section 5 Footnotes 2,3,4,5) containing video samples from Lifelong and Offline Learning models for every dataset (lower quality videos viewable online [here](https://drive.google.com/drive/folders/1nlc1NZG8lFZIE7w0hYsa2Np-YXjKjSJl)). Some reviewers also missed the reference to the quantitative results for two more baselines in the appendix for every dataset (Section 5 Line 398, Appendix D). These oversights have led to lower scores and skepticism regarding our main claim—that lifelong learning of video diffusion models is not only feasible but can achieve performance comparable to offline training given the same number of gradient steps. We kindly ask that you visit these resources should you question the validity of our claim and re-evaluate your scores in light of these results.
>
> To substantiate our main claim even further, we have incorporated new results on a real-world driving dataset despite operating within the limits of resources available to an academic lab. Please refer to the rebuttal addressed to Reviewer KLJW for the experiment setup and qualitative and quantitative results. These results support the validity of our claim on complex real-world video data.
>
> We now address specific questions and concerns.
> > The t in Equation (3) for lifelong learning exactly matches the total duration of the training video. Does i in Equation (2) cover the entire duration of the training video despite being randomly selected? Ensuring this would make the comparison fair.
>
> Thank you for the great question. Equation (3)’s t matches the number of training frames observed so far by the model. Equation (2)’s i covers the entire duration of the training video in our experiments to ensure that the comparison is fair. We will clarify this in the camera-ready draft.
> > If the training steps exceed the actual t in the training video, there is no difference between the lifelong and offline losses. How does the comparison between the two methods hold under these much longer training steps?
>
> Our lifelong learning setup requires at least one minibatch index at every timestep to be a real-time video frame, Lifelong Learning and Offline Learning cannot be compared once we have sequentially performed a single gradient step for all sliding windows of the video stream.
> > Will the same outcome apply for larger video U-Nets?
>
> While larger models could not be evaluated in this preliminary investigation due to compute restrictions, given Deepmind’s recent work [1] that shows that online learned discriminative models with 8 and 350 million parameters can match IID-trained model performance, we expect the same outcome to apply to larger video diffusion models. We note that our diffusion models have the same parameter count as the original paper's models [2]. Regardless, we have shown that lifelong learning can achieve a performance comparable to offline learning across a wide range of datasets.
> > Recent papers demonstrate that offline training for autoregressive diffusion can generate robust infinite single-game simulations for complex datasets. These works are missing from the paper.
>
> We believe that lifelong learning of advanced diffusion models capable of robustly generating long videos is an exciting next step and have included these references in the updated paper. We note that even the largest of these models, Genie 2, only stably generates videos up to a minute [3].
>
> ---
>
> We appreciate that the reviewer believes that the lifelong learning of video diffusion models is novel and can make a valuable contribution to the literature. We hope that our qualitative results that highlight indistinguishableness of lifelong and offline learning samples and comparison fairness clarifications will help the reviewer share Reviewer YRxK’s sentiment that our results strongly support the claim that training autoregressive video diffusion models from a single video stream is not only possible but can also be as effective as offline training given the same number of gradient steps.
>
> [1] Carreira, J., King, M., Patraucean, V., Gokay, D., Ionescu, C., Yang, Y., Zoran, D., Heyward, J., Doersch, C., Aytar, Y., Damen, D., and Zisserman, A. Learning from one continuous video stream, 2024.
>
> [2] Harvey, W., Naderiparizi, S., Masrani, V., Weilbach, C., & Wood, F. (2022). Flexible diffusion modeling of long videos. Advances in Neural Information Processing Systems, 35, 27953-27965.
>
> [3] Genie 2: A large-scale foundation world model. (2025, March 25). Google DeepMind. deepmind.google/discover/blog/genie-2-a-large-scale-foundation-world-model/

---

> > ### Comment · Reviewer_gLa1 · 2025-04-06
> >
> > Thanks for the rebuttals. However, comparing models under limited training steps isn’t a fair evaluation. In most cases, the offline model can be trained for more steps. The scalability is still unproven, so I’ve decided to keep my original rating.

---

### Decision · Program_Chairs · 2025-05-01

**Decision:**

Reject

**Comment:**

This paper proposes, for the first time, learning a video diffusion model in a lifelong learning setting from a single video stream. It provides a proof-of-concept using three synthetic datasets, showing that the lifelong learned model with a replay buffer achieves comparable results to an IID-trained model with the same number of gradient steps.

While reviewers appreciate the novel problem setting and find the results interesting, the majority consider the empirical evaluation inadequate. In particular, the experiments are limited to a few synthetic datasets with a small model size, lack comparison with recent video diffusion models, do not sufficiently analyze lifelong learning-specific characteristics, and the comparison under a limited number of gradient steps is not fair to offline-trained models.

The authors' rebuttal offers some justification for their experimental setup and adds a new real-world experiment, but it does not sufficiently address the reviewers' various concerns.